# The Population Posterior
# and Bayesian Modeling on Streams

**James McInerney**
Columbia University
james@cs.columbia.edu

**Rajesh Ranganath**
Princeton University
rajeshr@cs.princeton.edu

**David Blei**
Columbia University
david.blei@columbia.edu

## Abstract

Many modern data analysis problems involve inferences from streaming data. However, streaming data is not easily amenable to the standard probabilistic modeling approaches, which require conditioning on finite data. We develop population variational Bayes, a new approach for using Bayesian modeling to analyze streams of data. It approximates a new type of distribution, the population posterior, which combines the notion of a population distribution of the data with Bayesian inference in a probabilistic model. We develop the population posterior for latent Dirichlet allocation and Dirichlet process mixtures. We study our method with several large-scale data sets.

## 1 Introduction

Probabilistic modeling has emerged as a powerful tool for data analysis. It is an intuitive language for describing assumptions about data and provides efficient algorithms for analyzing real data under those assumptions. The main idea comes from Bayesian statistics. We encode our assumptions about the data in a structured probability model of hidden and observed variables; we condition on a data set to reveal the posterior distribution of the hidden variables; and we use the resulting posterior as needed, for example to form predictions through the posterior predictive distribution or to explore the data through the posterior expectations of the hidden variables.

Many modern data analysis problems involve inferences from streaming data. Examples include exploring the content of massive social media streams (e.g., Twitter, Facebook), analyzing live video streams, estimating the preferences of users on an online platform for recommending new items, and predicting human mobility patterns for anticipatory computing. Such problems, however, cannot easily take advantage of the standard approach to probabilistic modeling, which requires that we condition on a finite data set.

This might be surprising to some readers; after all, one of the tenets of the Bayesian paradigm is that we can update our posterior when given new information. ("Yesterday's posterior is today's prior.") But there are two problems with using Bayesian updating on data streams. The first problem is that Bayesian inference computes posterior uncertainty under the assumption that the model is correct. In theory this is sensible, but only in the impossible scenario where the data truly came from the proposed model. In practice, all models provide approximations to the data-generating distribution, and when the model is incorrect, the uncertainty that maximizes predictive likelihood may be larger or smaller than the Bayesian posterior variance. This problem is exacerbated in potentially never-ending streams; after seeing only a few data points, uncertainty is high, but eventually the model becomes overconfident.

The second problem is that the data stream might change over time. This is an issue because, frequently, our goal in applying probabilistic models to streams is not to characterize how they change, but rather to accommodate it. That is, we would like for our current estimate of the latent variables to be accurate to the current state of the stream and to adapt to how the stream might slowly

change. (This is in contrast, for example, to time series modeling.) Traditional Bayesian updating cannot handle this. Either we explicitly model the time series, and pay a heavy inferential cost, or we tacitly assume that the data are exchangeable, i.e., that the underlying distribution does not change.

In this paper we develop new ideas for analyzing data streams with probabilistic models. Our approach combines the frequentist notion of the population distribution with probabilistic models and Bayesian inference.

**Main idea: The population posterior.** Consider a latent variable model of $\alpha$ data points. (This is unconventional notation; we will describe why we use it below.) Following [14], we define the model to have two kinds of hidden variables: global hidden variables $\beta$ contain latent structure that potentially governs any data point; local hidden variables $z_i$ contain latent structure that only governs the $i$th data point. Such models are defined by the joint,

$$p(\beta, \mathbf{z}, \mathbf{x}) = p(\beta) \prod_{i=1}^{\alpha} p(x_i, z_i \mid \beta), \tag{1}$$

where $\mathbf{x} = x_{1:\alpha}$ and $\mathbf{z} = z_{1:\alpha}$. Traditional Bayesian statistics conditions on a fixed data set $\mathbf{x}$ to obtain the posterior distribution of the hidden variables $p(\beta, \mathbf{z} \mid \mathbf{x})$. As we discussed, this framework cannot accommodate data streams. We need a different way to use the model.

We define a new distribution, the *population posterior*, which enables us to consider Bayesian modeling of streams. Suppose we observe $\alpha$ data points independently from the underlying population distribution, $\mathbf{X} \sim F_\alpha$. This induces a posterior $p(\beta, \mathbf{z} \mid \mathbf{X})$, which is a function of the random data. The population posterior is the expected value of this distribution,

$$\mathbb{E}_{F_\alpha}\left[p(\mathbf{z}, \beta \mid \mathbf{X})\right] = \mathbb{E}_{F_\alpha}\left[\frac{p(\beta, \mathbf{z}, \mathbf{X})}{p(\mathbf{X})}\right]. \tag{2}$$

Notice that this distribution is not a function of observed data; it is a function of the population distribution $F$ and the data size $\alpha$. The data size is a hyperparameter that can be set; it effectively controls the variance of the population posterior. How to best set it depends on how close the model is to the true data distribution.

We have defined a new problem. Given an endless stream of data points coming from $F$ and a value for $\alpha$, our goal is to approximate the corresponding population posterior. In this paper, we will approximate it through an algorithm based on variational inference and stochastic optimization. As we will show, our algorithm justifies applying a variant of stochastic variational inference [14] to a data stream. We used our method to analyze several data streams with two modern probabilistic models, latent Dirichlet allocation [5] and Dirichlet process mixtures [11]. With held-out likelihood as a measure of model fitness, we found our method to give better models of the data than approaches based on full Bayesian inference [14] or Bayesian updating [8].

**Related work.** Researchers have proposed several methods for inference on streams of data. Refs. [1, 9, 27] propose extending Markov chain Monte Carlo methods for streaming data. However, sampling-based approaches do not scale to massive datasets; the variational approximation enables more scalable inference. In variational inference, Ref. [15] propose online variational inference by exponentially forgetting the variational parameters associated with old data. Stochastic variational inference (SVI) [14] also decay parameters derived from old data, but interprets this in the context of stochastic optimization. Neither of these methods applies to streaming data; both implicitly rely on the data being of known size (even when subsampling data to obtain noisy gradients).

To apply the variational approximation to streaming data, Ref. [8] and Ref. [12] both propose Bayesian updating of the approximating family; Ref. [22] adapts this framework to nonparametric mixture models. Here we take a different approach, changing the variational objective to incorporate a population distribution and then following stochastic gradients of this new objective. In Section 3 we show that this generally performs better than Bayesian updating.

Independently, Ref. [23] applied SVI to streaming data by accumulating new data points into a growing window and then uniformly sampling from this window to update the variational parameters. Our method justifies that approach. Further, they propose updating parameters along a trust region, instead of following (natural) gradients, as a way of mitigating local optima. This innovation can be incorporated into our method.

## 2 Variational Inference for the Population Posterior

We develop *population variational Bayes*, a method for approximating the population posterior in Eq. 2. Our method is based on variational inference and stochastic optimization.

**The F-ELBO.** The idea behind variational inference is to approximate difficult-to-compute distributions through optimization [16, 25]. We introduce an approximating family of distributions over the latent variables $q(\beta, \mathbf{z})$ and try to find the member of $q(\cdot)$ that minimizes the Kullback-Leibler (KL) divergence to the target distribution.

Population variational Bayes (VB) uses variational inference to approximate the population posterior in Eq. 2. It aims to minimize the KL divergence from an approximating family,

$$q^*(\beta, \mathbf{z}) = \arg\min_q \mathrm{KL}(q(\beta, \mathbf{z}) || \mathbb{E}_{F_\alpha}[p(\beta, \mathbf{z} | \mathbf{X})]). \tag{3}$$

As for the population posterior, this objective is a function of the population distribution of $\alpha$ data points $F_\alpha$. Notice the difference to classical VB. In classical VB, we optimize the KL divergence between $q(\cdot)$ and a posterior, $\mathrm{KL}(q(\beta, \mathbf{z}) || p(\beta, \mathbf{z} | \mathbf{x})$; its objective is a function of a fixed data set $\mathbf{x}$. In contrast, the objective in Eq. 3 is a function of the population distribution $F_\alpha$.

We will use the mean-field variational family, where each latent variable is independent and governed by a free parameter,

$$q(\beta, \mathbf{z}) = q(\beta | \lambda) \prod_{i=1}^{\alpha} q(z_i | \phi_i). \tag{4}$$

The free variational parameters are the global parameters $\lambda$ and local parameters $\phi_i$. Though we focus on the mean-field family, extensions could consider structured families [13, 20], where there is dependence between variables.

In classical VB, where we approximate the usual posterior, we cannot compute the KL. Thus, we optimize a proxy objective called the ELBO (evidence lower bound) that is equal to the negative KL up to an additive constant. Maximizing the ELBO is equivalent to minimizing the KL divergence to the posterior.

In population VB we also optimize a proxy objective, the F-ELBO. The F-ELBO is an expectation of the ELBO under the population distribution of the data,

$$\mathscr{L}(\lambda, \phi; F_\alpha) = \mathbb{E}_{F_\alpha} \left[ \mathbb{E}_q \left[ \log p(\beta) - \log q(\beta | \lambda) + \sum_{i=1}^{\alpha} \log p(X_i, Z_i | \beta) - \log q(Z_i) \right] \right]. \tag{5}$$

The F-ELBO is a lower bound on the population evidence $\log \mathbb{E}_{F_\alpha}[p(\mathbf{X})]$ and a lower bound on the negative KL to the population posterior. (See Appendix A.) The inner expectation is over the latent variables $\beta$ and $\mathbf{Z}$, and is a function of the variational distribution $q(\cdot)$. The outer expectation is over the $\alpha$ random data points $\mathbf{X}$, and is a function of the population distribution $F_\alpha(\cdot)$. The F-ELBO is thus a function of both the variational distribution and the population distribution.

As we mentioned, classical VB maximizes the (classical) ELBO, which is equivalent to minimizing the KL. The F-ELBO, in contrast, is only a bound on the negative KL to the population posterior. Thus maximizing the F-ELBO is suggestive but is not guaranteed to minimize the KL. That said, our studies show that this is a good quantity to optimize, and in Appendix A we show that the F-ELBO does minimize $\mathbb{E}_{F_\alpha}[\mathrm{KL}(q(\mathbf{z}) || p(\mathbf{z}, \beta | \mathbf{X}))]$, the population KL.

**Conditionally conjugate models.** In the next section we will develop a stochastic optimization algorithm to maximize Eq. 5. First, we describe the class of models that we will work with.

Following [14] we focus on conditionally conjugate models. A conditionally conjugate model is one where each complete conditional—the conditional distribution of a latent variable given all the other latent variables and the observations—is in the exponential family. This class includes many models in modern machine learning, such as mixture models, topic models, many Bayesian nonparametric models, and some hierarchical regression models. Using conditionally conjugate models simplifies many calculations in variational inference.

Under the joint in Eq. 1, we can write a conditionally conjugate model with two exponential families:

$$p(z_i, x_i \,|\, \beta) = h(z_i, x_i) \exp\left\{\beta^\top t(z_i, x_i) - a(\beta)\right\} \tag{6}$$

$$p(\beta \,|\, \zeta) = h(\beta) \exp\left\{\zeta^\top t(\beta) - a(\zeta)\right\}. \tag{7}$$

We overload notation for base measures $h(\cdot)$, sufficient statistics $t(\cdot)$, and log normalizers $a(\cdot)$. Note that $\zeta$ is the hyperparameter and that $t(\beta) = [\beta, -a(\beta)]$ [3].

In conditionally conjugate models each complete conditional is in an exponential family, and we use these families as the factors in the variational distribution in Eq. 4. Thus $\lambda$ indexes the same family as $p(\beta \,|\, \mathbf{z}, \mathbf{x})$ and $\phi_i$ indexes the same family as $p(z_i \,|\, x_i, \beta)$. For example, in latent Dirichlet allocation [5], the complete conditional of the topics is a Dirichlet; the complete conditional of the per-document topic mixture is a Dirichlet; and the complete conditional of the per-word topic assignment is a categorical. (See [14] for details.)

**Population variational Bayes.** We have described the ingredients of our problem. We are given a conditionally conjugate model, described in Eqs. 6 and 7, a parameterized variational family in Eq. 4, and a stream of data from an unknown population distribution $F$. Our goal is to optimize the F-ELBO in Eq. 5 with respect to the variational parameters.

The F-ELBO is a function of the population distribution, which is an unknown quantity. To overcome this hurdle, we will use the stream of data from $F$ to form noisy gradients of the F-ELBO; we then update the variational parameters with stochastic optimization (a technique to find a local optimum by following noisy unbiased gradients [7]).

Before describing the algorithm, however, we acknowledge one technical detail. Mirroring [14], we optimize an F-ELBO that is only a function of the global variational parameters. The one-parameter population VI objective is $\mathscr{L}_{F_\alpha}(\lambda) = \max_\phi \mathscr{L}_{F_\alpha}(\lambda, \phi)$. This implicitly optimizes the local parameter as a function of the global parameter and allows us to convert the potentially infinite-dimensional optimization problem in Eq. 5 to a finite one. The resulting objective is identical to Eq. 5, but with $\phi$ replaced by $\phi(\lambda)$. (Details are in Appendix B).

The next step is to form a noisy gradient of the F-ELBO so that we can use stochastic optimization to maximize it. Stochastic optimization maximizes an objective by following noisy and unbiased gradients [7, 19]. We will write the gradient of the F-ELBO as an expectation with respect to $F_\alpha$, and then use Monte Carlo estimates to form noisy gradients.

We compute the gradient of the F-ELBO by bringing the gradient operator inside the expectations of Eq. 5.[1] This results in a population expectation of the classical VB gradient with $\alpha$ data points.

We take the natural gradient [2], which has a simple form in completely conjugate models [14]. Specifically, the natural gradient of the F-ELBO is

$$\hat{\nabla}_\lambda \mathscr{L}(\lambda; F_\alpha) = \zeta - \lambda + \mathbb{E}_{F_\alpha}\left[\sum_{i=1}^{\alpha} \mathbb{E}_{\phi_i(\lambda)}\left[t(x_i, Z_i)\right]\right]. \tag{8}$$

We approximate this expression using Monte Carlo to compute noisy, unbiased natural gradients at $\lambda$. To form the Monte Carlo estimate, we collect $\alpha$ data points from $F$; for each we compute the optimal local parameters $\phi_i(\lambda)$, which is a function of the sampled data point and variational parameters; we then compute the quantity inside the brackets in Eq. 8. Averaging these results gives the Monte Carlo estimate of the natural gradient. We follow the noisy natural gradient and repeat.

The algorithm is summarized in Algorithm 1. Because Eq. 8 is a Monte Carlo estimate, we are free to draw $B$ data points from $F_\alpha$ (where $B << \alpha$) and rescale the sufficient statistics by $\alpha/B$. This makes the natural gradient estimate noisier, but faster to calculate. As highlighted in [14], this strategy is more computationally efficient because early iterations of the algorithm have inaccurate values of $\lambda$. It is wasteful to pass through a lot of data before making updates to $\lambda$.

**Discussion.** Thus far, we have defined the population posterior and showed how to approximate it with population variational inference. Our derivation justifies using an algorithm like stochastic variational inference (SVI) [14] on a stream of data. It is nearly identical to SVI, but includes an additional parameter: the number of data points in the population posterior $\alpha$.

**Algorithm 1** Population Variational Bayes

---

Randomly initialize global variational parameter $\lambda^{(0)}$
Set iteration $t \leftarrow 0$
**repeat**
    Draw data minibatch $x_{1:B} \sim F_\alpha$
    Optimize local variational parameters $\phi_1(\lambda^{(t)}), \ldots, \phi_B(\lambda^{(t)})$
    Calculate natural gradient $\hat{\nabla}_\lambda \mathscr{L}(\lambda^{(t)}; F_\alpha)$ [see Eq. 8]
    Update global variational parameter with learning rate $\rho^{(t)}$
        $\lambda^{(t+1)} = \lambda^{(t)} + \rho^{(t)} \frac{\alpha}{B} \hat{\nabla}_\lambda \mathscr{L}(\lambda^{(t)}; F_\alpha)$
    Update iteration count $t \leftarrow t + 1$
**until** forever

---

Note we can recover the original SVI algorithm as an instance of population VI, thus reinterpreting it as minimizing the KL divergence to the population posterior. We recover SVI by setting $\alpha$ equal to the number of data points in the data set and replacing the stream of data $F$ with $\hat{F}_\mathbf{x}$, the empirical distribution of the observations. The "stream" in this case comes from sampling with replacement from $\hat{F}_\mathbf{x}$, which results in precisely the original SVI algorithm.[2]

We focused on the conditionally conjugate family for convenience, i.e., the simple gradient in Eq. 8. We emphasize, however, that by using recent tools for nonconjugate inference [17, 18, 24], we can adapt the new ideas described above—the population posterior and the F-ELBO—outside of conditionally conjugate models.

Finally, we analyze the population posterior distribution under the assumption the only way the stream affects the model is through the data. Formally, this means the unobserved variables in the model and the stream $F_\alpha$ are independent given the data $\mathbf{X}$. The population posterior without the local latent variables $\mathbf{z}$ (which can be marginalized out) is $\mathbb{E}_{F_\alpha}[p(\beta \mid \mathbf{X})]$. Expanding the expectation gives $\int p(\beta \mid \mathbf{X}) p(\mathbf{X} \mid F_\alpha) d\mathbf{X}$, showing that the population posterior distribution can be written as $p(\beta \mid F_\alpha)$. This can be depicted as a graphical model:

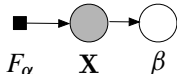

$$F_\alpha \quad \mathbf{X} \quad \beta$$

This means first, that the population posterior is well defined even when the model does not specify the marginal distribution of the data and, second, rather than the classical Bayesian setting where the posterior is conditioned on a finite fixed dataset, the population posterior is a distributional posterior conditioned on the stream $F_\alpha$.

## 3 Empirical Evaluation

We study the performance of population variational Bayes (population VB) against SVI and streaming variational Bayes (SVB) [8]. With large real-world data we study two models, latent Dirichlet allocation [5] and Bayesian nonparametric mixture models, comparing the held-out predictive performance of the algorithms. All three methods share the same local variational update, which is the dominating computational cost. We study the data coming in a true ordered stream, and in a permuted stream (to better match the assumptions of SVI). Across data and models, population VB usually outperforms the existing approaches.

**Models.** We study two models. The first is latent Dirichlet allocation (LDA) [5]. LDA is a mixed-membership model of text collections and is frequently used to find its latent topics. LDA assumes that there are $K$ topics $\beta_k \sim \text{Dir}(\eta)$, each of which is a multinomial distribution over a fixed vocabulary. Documents are drawn by first choosing a distribution over topics $\theta_d \sim \text{Dir}(\alpha)$ and then

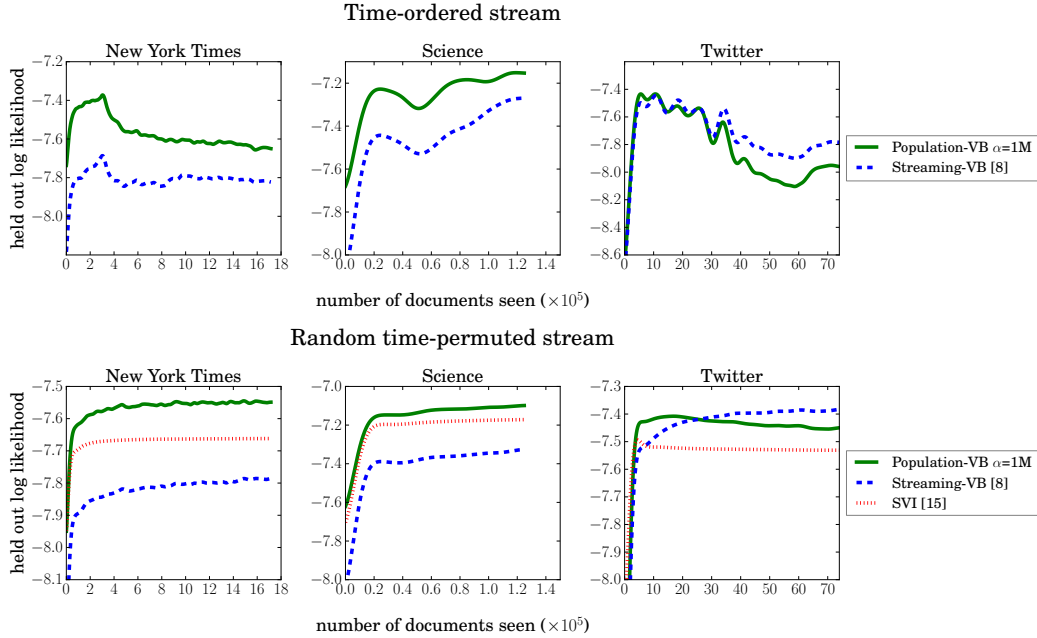

**Figure 1:** Held out predictive log likelihood for LDA on large-scale streamed text corpora. Population-VB outperforms existing methods for two out of the three settings. We use the best settings of $\alpha$.

drawing each word by choosing a topic assignment $z_{dn} \sim \text{Mult}(\theta_d)$ and finally choosing a word from the corresponding topic $w_{dn} \sim \beta_{z_{dn}}$. The joint distribution is

$$p(\beta,\theta,\mathbf{z},w|\eta,\gamma) = p(\beta|\eta) \prod_{d=1}^{\alpha} p(\theta_d|\gamma) \prod_{i=1}^{N} p(z_{di}|\theta_d)p(w_{di}|\beta,z_{di}). \tag{9}$$

Fixing hyperparameters, the inference problem is to estimate the conditional distribution of the topics given a large collection of documents.

The second model is a Dirichlet process (DP) mixture [11]. Loosely, DP mixtures are mixture models with a potentially infinite number of components; thus choosing the number of components is part of the posterior inference problem. When using variational inference for DP mixtures [4], we take advantage of the stick breaking representation to construct a truncated variational approximation [21]. The variables are mixture proportions $\pi \sim \text{Stick}(\eta)$, mixture components $\beta_k \sim H(\gamma)$ (for infinite $k$), mixture assignments $z_i \sim \text{Mult}(\pi)$, and observations $x_i \sim G(\beta_{z_i})$. The joint is

$$p(\beta,\pi,\mathbf{z},\mathbf{x}|\eta,\gamma) = p(\pi|\eta)p(\beta|\gamma) \prod_{i=1}^{\alpha} p(z_i|\pi)p(x_i|\beta,x_i). \tag{10}$$

The likelihood and prior on the components are general to the observations at hand. In our study of real-valued data we use normal priors and normal likelihoods; in our study of text data we use Dirichlet priors and multinomial likelihoods.

For both models we vary $\alpha$, usually fixed to the number of data points in traditional analysis.

**Datasets.** With LDA we analyze three large-scale streamed corpora: 1.7M articles from the New York Times spanning 10 years, 130K Science articles written over 100 years, and 7.4M tweets collected from Twitter on Feb 2nd, 2014. We processed them all in a similar way, choosing a vocabulary based on the most frequent words in the corpus (with stop words removed): 8,000 for the New York Times, 5,855 for Science, and 13,996 for Twitter. On Twitter, each tweet is a document, and we removed duplicate tweets and tweets that did not contain at least 2 words in the vocabulary. For each data stream, all algorithms took a few hours to process all the examples we collected.

With DP mixtures, we analyze human location behavior data. These data allow us to build periodic models of human population mobility, with applications to disaster response and urban planning. Such models account for periodicity by including the hour of the week as one of the dimensions of the

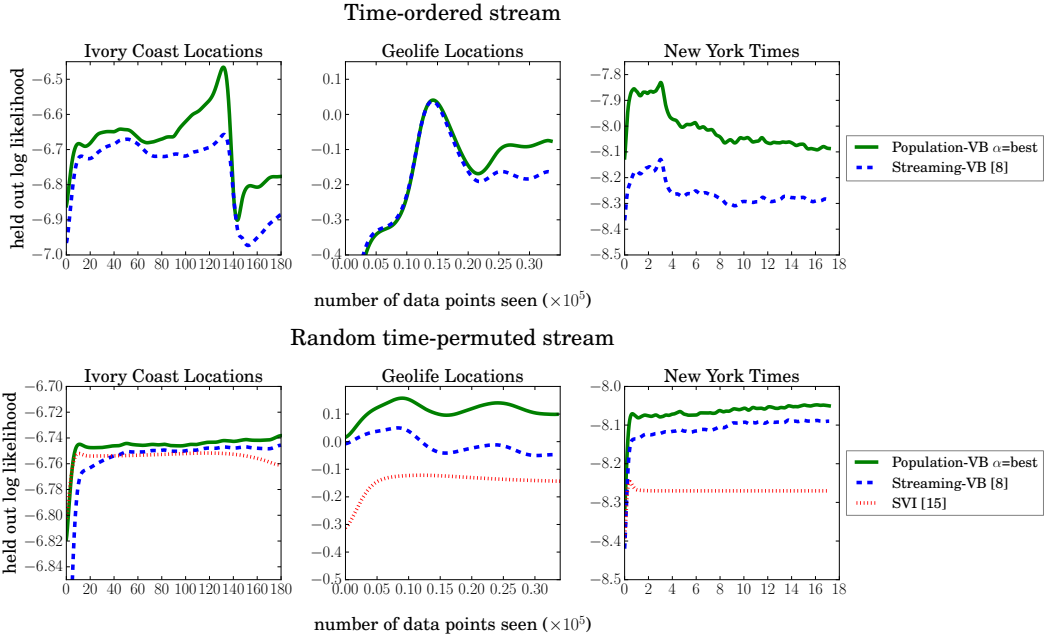

**Figure 2:** Held out predictive log likelihood for Dirichlet process mixture models on large-scale streamed location and text data sets. Note that we apply Gaussian likelihoods in the Geolife dataset, so the reported predictive performance is measured by probability density. We chose the best $\alpha$ for each population-VB curve.

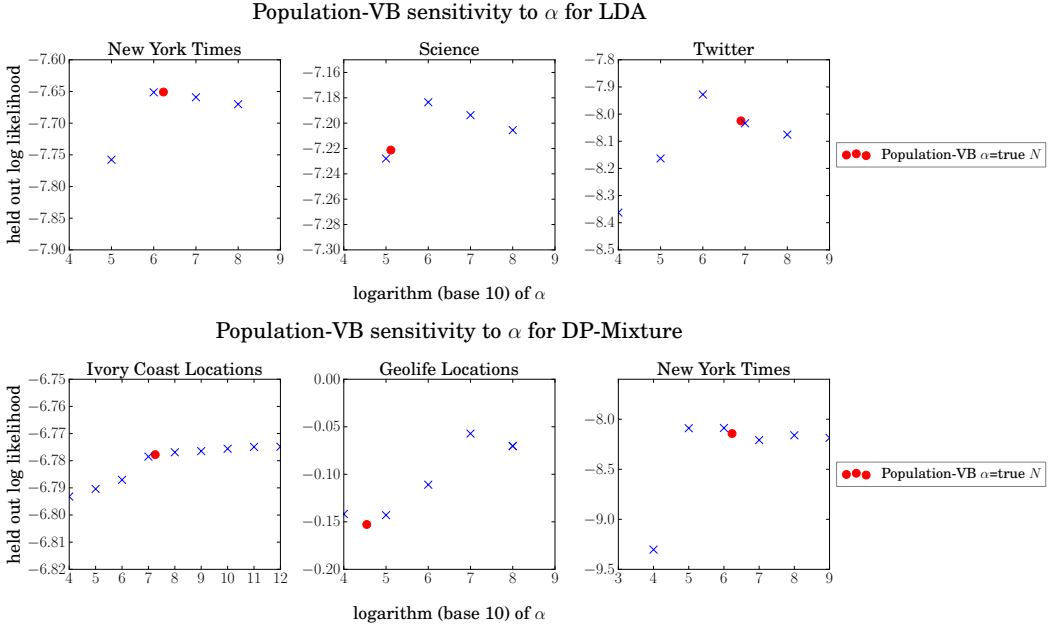

**Figure 3:** We show the sensitivity of population-VB to hyperparameter $\alpha$ (based on final log likelihoods in the time-ordered stream) and find that the best setting of $\alpha$ often differs from the true number of data points (which may not be known in any case in practice).

data to be modeled. The Ivory Coast location data contains 18M discrete cell tower locations for 500K users recorded over 6 months [6]. The Microsoft Geolife dataset contains 35K latitude-longitude GPS locations for 182 users over 5 years. For both data sets, our observations reflect down-sampling the data to ensure that each individual is seen no more than once every 15 minutes.

**Results.** We compare population VB with SVI [14] and SVB [8] for LDA [8] and DP mixtures [22]. SVB updates the variational approximation of the global parameter using density filtering with exponential families. The complexity of the approximation remains fixed as the expected sufficient statistics from minibatches observed in a stream are combined with those of the current approximation. (Here we give the final results. We include details of how we set and fit hyperparameters below.)

We measure model fitness by evaluating the average predictive log likelihood on held-out data. This involves splitting held-out observations (that were not involved in the posterior approximation of $\beta$) into two equal halves, inferring the local component distribution based on the first half, and testing with the second half [14, 26]. For DP-mixtures, we condition on the observed hour of the week and predict the geographic location of the held-out data point.

In standard offline studies, the held-out set is randomly selected from the data. With streams, however, we test on the next 10K documents (for New York Times, Science), 500K tweets (for Twitter), or 25K locations (on Geo data). This is a valid held-out set because the data ahead of the current position in the stream have not yet been seen by the inference algorithms.

Figure 1 shows the performance for LDA. We looked at two types of streams: one in which the data appear in order and the other in which they have been permuted (i.e., an exchangeable stream). The time permuted stream reveals performance when each data minibatch is safely assumed to be an i.i.d. sample from $F$; this results in smoother improvements to predictive likelihood. On our data, we found that population VB outperformed SVI and SVB on two of the data sets and outperformed SVI on all of the data. SVB performed better than population VB on Twitter.

Figure 2 shows a similar study for DP mixtures. We analyzed the human mobility data and the New York Times. (Ref. [22] also analyzed the New York Times.) On these data population VB outperformed SVB and SVI in all settings.[3]

**Hyperparameters** Unlike traditional Bayesian methods, the data set size $\alpha$ is a hyperparameter to population VB. It helps control the posterior variance of the population posterior. Figure 3 reports sensitivity to $\alpha$ for all studies (for the time-ordered stream). These plots indicate that the optimal setting of $\alpha$ is often different from the true number of data points; the best performing population posterior variance is not necessarily the one implied by the data. The other hyperparameters to our experiments are reported in Appendix C.

## 4   Conclusions and Future Work

We introduced the population posterior, a distribution over latent variables that combines traditional Bayesian inference with the frequentist idea of the population distribution. With this idea, we derived population variational Bayes, an efficient algorithm for probabilistic inference on streams. On two complex Bayesian models and several large data sets, we found that population variational Bayes usually performs better than existing approaches to streaming inference.

In this paper, we made no assumptions about the structure of the population distribution. Making assumptions, such as the ability to obtain streams conditional on queries, can lead to variants of our algorithm that learn which data points to see next during inference. Finally, understanding the theoretical properties of the population posterior is also an avenue of interest.

**Acknowledgments.** We thank Allison Chaney, John Cunningham, Alp Kucukelbir, Stephan Mandt, Peter Orbanz, Theo Weber, Frank Wood, and the anonymous reviewers for their comments. This work is supported by NSF IIS-0745520, IIS-1247664, IIS-1009542, ONR N00014-11-1-0651, DARPA FA8750-14-2-0009, N66001-15-C-4032, NDSEG, Facebook, Adobe, Amazon, and the Siebel Scholar and John Templeton Foundations.

## Footnotes

[1] For most models of interest, this is justified by the dominated convergence theorem.

[2]This derivation of SVI is an application of Efron's plug-in principle [10] applied to inference of the population posterior. The plug-in principle says that we can replace the population $F$ with the empirical distribution of the data $\hat{F}$ to make population inferences. In our empirical study, however, we found that population VI often outperforms stochastic VI. Treating the data in a true stream, and setting the number of data points different to the true number, can improve predictive accuracy.

[3]Though our purpose is to compare algorithms, we make one note about a specific data set. The predictive accuracy for the Ivory Coast data set plummets after 14M data points. This is because of the data collection policy. For privacy reasons the data set provides the cell tower locations of a randomly selected cohort of 50K users every 2 weeks [6]. The new cohort at 14M data points behaves differently to previous cohorts in a way that affects predictive performance. However, both algorithms steadily improve after this shock.

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
