[Supplementary Material · appendices.pdf]

# Appendix for the Population Posterior
# and Bayesian Modeling on Streams

**James McInerney**
Columbia University
james@cs.columbia.edu

**Rajesh Ranganath**
Princeton University
rajeshr@cs.princeton.edu

**David Blei**
Columbia University
david.blei@columbia.edu

## A   Derivation of the F-ELBO

Classic variational inference seeks to minimize $\mathrm{KL}(q(\beta,\mathbf{z})||p(\beta,\mathbf{z}\,|\,\mathbf{x}))$ using the following equivalence to show that the negative evidence lower bound (ELBO) is an appropriate surrogate objective to be minimized,

$$\log p(\mathbf{x}) \quad = \quad \mathrm{KL}(q(\beta,\mathbf{z})||p(\mathbf{z}\,|\,\mathbf{x})) + \mathbb{E}_q[\log p(\beta,\mathbf{z},\mathbf{x}) - \log q(\beta,\mathbf{z})]. \tag{1}$$

This equivalence arises from the definition of KL divergence [6].

To derive the F-ELBO, replace $\mathbf{x}$ with a draw $\mathbf{X}$ of size $\alpha$ from the population distribution, $\mathbf{X} \sim F_\alpha$, then apply an expectation with respect to $F_\alpha$ to both sides of Eq.1,

$$\begin{aligned}
\mathbb{E}_{F_\alpha}[\log p(\mathbf{X})] \quad &= \quad \mathbb{E}_{F_\alpha}[\mathrm{KL}(q(\beta,\mathbf{z})||p(\beta,\mathbf{z}\,|\,\mathbf{X})) + \mathbb{E}_q[\log p(\beta,\mathbf{z},\mathbf{X}) - \log q(\beta,\mathbf{z})]] \\
&= \quad \mathbb{E}_{F_\alpha}[\mathrm{KL}(q(\beta,\mathbf{z})||p(\beta,\mathbf{z}\,|\,\mathbf{X}))] + \mathbb{E}_{F_\alpha}[\mathbb{E}_q[\log p(\beta,\mathbf{z},\mathbf{X}) - \log q(\beta,\mathbf{z})]]. \quad (2)
\end{aligned}$$

This confirms that the negative F-ELBO is a surrogate objective for $\mathbb{E}_{F_\alpha}[\mathrm{KL}(q(\beta,\mathbf{z})||p(\beta,\mathbf{z}\,|\,\mathbf{X}))]$ because $q(\cdot)$ does not appear on the left hand side of Eq. 2.

Now use the fact that logarithm is a concave function and apply Jensen's inequality to Eq. 2 to show that the F-ELBO is a lower bound on the population evidence,

$$\begin{aligned}
\mathbb{E}_{F_\alpha}[\mathbb{E}_q[\log p(\beta,\mathbf{z},\mathbf{X}) - \log q(\beta,\mathbf{z})]] \quad &\leq \quad \mathbb{E}_{F_\alpha}[\log p(\mathbf{X})] \\
&\leq \quad \log \mathbb{E}_{F_\alpha}[p(\mathbf{X})]. \quad (3)
\end{aligned}$$

Additionally, Jensen's inequality applied to Eq. 2 in a different way shows that maximizing the F-ELBO minimizes an upper bound on the KL divergence between $q(\cdot)$ and the population posterior,

$$\begin{aligned}
\mathbb{E}_{F_\alpha}[\mathrm{KL}(q(\beta,\mathbf{z})||p(\beta,\mathbf{z}\,|\,\mathbf{X}))] \quad &= \quad \mathbb{E}_q[\log q(\beta,\mathbf{z})] - \mathbb{E}_{F_\alpha}[\mathbb{E}_q[\log p(\beta,\mathbf{z}\,|\,\mathbf{X})]] \\
&\geq \quad \mathbb{E}_q[\log q(\beta,\mathbf{z})] - \mathbb{E}_q[\log \mathbb{E}_{F_\alpha}[p(\beta,\mathbf{z}\,|\,\mathbf{X})]] \\
&= \quad \mathrm{KL}(q(\beta,\mathbf{z})||\mathbb{E}_{F_\alpha}[p(\beta,\mathbf{z}\,|\,\mathbf{X})]), \quad (4)
\end{aligned}$$

where we have exchanged expectations with respect to $q(\cdot)$ and $F_\alpha$.

## B   One-Parameter F-ELBO

The F-ELBO for conditionally conjugate exponential families is as follows

$$\mathscr{L}(\lambda,\phi;F_\alpha) = \mathbb{E}_{F_\alpha}\left[\mathbb{E}_q\left[\log p(\beta) - \log q(\beta\,|\,\lambda) + \sum_{i=1}^{\alpha}\log p(X_i,Z_i\,|\,\beta) - \log q(Z_i)\right]\right].$$

This can be rewritten in terms of just the global variational parameters. We define the one parameter population variational inference objective as $\mathscr{L}_{F_\alpha}(\lambda) = \max_\phi \mathscr{L}_{F_\alpha}(\lambda,\phi)$. We can write this more

compactly if we let $\phi_i(\lambda)$ be the value of $\phi_i$ that maximizes the F-ELBO given $\lambda$.[1] Formally, this gives

$$\mathcal{L}_{F_\alpha}(\lambda) = \mathbb{E}_{q(\beta\,|\,\lambda)} \left[ \log p(\beta) - \log q(\beta\,|\,\lambda) + \mathbb{E}_{\mathbf{X}\sim F_\alpha} [\sum_{i=1}^{\alpha} \mathbb{E}_{q(Z_i\,|\,\phi_i(\lambda))} [\log p(X_i, Z_i\,|\,\beta) - \log q(Z_i)]] \right],$$

where we have moved the expectation with respect to $F_\alpha$ inside the expectation with respect to $q(\cdot)$.

## C   Hyperparameters for Experiments

Our methods are based on stochastic optimization and require setting the learning rate [4]. For all gradient-based procedures, we used a small fixed learning rate to follow noisy gradients. We note that adaptive learning rates [1, 5] are also applicable in this setting, though we did not observe an improvement using these for time-ordered streams.

Our procedures also require setting a batch size, how many data points we observe before updating the approximate posterior. In the LDA study we set the batch size to 100 documents for the larger corpora (New York Times, Science) and 5,000 for Twitter. These sizes were selected to make the average number of words per batch equal in both settings, which helps lower the variance of the gradients. In the DP mixture study we use a batch size of 5,000 locations for Ivory Coast, 500 locations for Geolife, and 100 documents for New York Times.

LDA requires additional hyperparameters. In line with Ref. [2], we used 100 topics, and set the hyperparameter to the global topics (which controls the sparsity of topics) to $\eta = 0.01$ and the hyperparameter to the word-topic assignments (which controls the sparsity of topic memberships) to $\gamma = 0.1$. The variational approximation for the DP mixture model requires a truncation hyperparameter $K$. We set it to 100 for all three data sets and verified that the inference algorithm used fewer than this number of components.

## Footnotes

[1]The optimal local variational parameter $\phi_i$ can be computed using gradient ascent or coordinate ascent as done in [3].