[Reviews · NeurIPS 2015]

Submitted by Assigned_Reviewer_1

Update after author feedback and discussion: I am disappointed in the feedback and the authors not commenting about the practicality of the method for actual streaming data, and thus decreasing my score. As far as I understand, the algorithm could resample any earlier points again and hence to apply it one would need to record and repeatedly access the entire stream. This is clearly not comparable with real streaming algorithms and this would need to be made more explicit.
Summary: Interesting suggestion to generalise Bayesian inference for infinite data streams. I would have liked to see more comparisons about how the problem could be approached in a Bayesian setting by changing the model, and did not see how this is streaming as you might in theory have to resample the very first data points again much later.

Submitted by Assigned_Reviewer_2

This paper proposed the population posterior distribution for Bayesian modeling of streams of data and showed how stochastic optimization could be used to find a good approximation.

The proposed framework and algorithm were demonstrated on both latent Dirichlet allocation and Dirichlet process mixture models on text and geolocation data and were shown to perform better than previous work in some cases.

Overall, I think the main idea of the paper is very interesting and it would fit in well at NIPS.

There are a few aspects of the paper that could use some more discussion though.

First, the authors were very careful throughout the paper to use the term "Bayesian modeling", except the title uses "Bayesian inference", which this paper definitely does not provide a method for.

The title should really use "Bayesian modeling" instead.

Also, the notation used in Eqs. 3 and 4 for the local variables is confusing as they're being optimized to the expectation of a population average.

However, they're local to a particular data point.

Perhaps there's a better way to write this because as written it looks like the learned local variational parameters will just be mess because they'll all be averaged together.

I see how everything works in the actual algorithm, I'm just hoping there's a clean way to write this in Eqs. 3 and 4.

Also, the step-size for gradient-ascent was never introduced in the algorithm.

Finally, in the paragraph around line 153, the authors say that optimizing the F-ELBO is not guaranteed to minimize the KL, but in the sentence immediately after they say they show that it does in Appendix A.

This needs to be explained better, because these sentences say opposite things.

A quick discussion about the \alpha parameter is given in the experiments, however, the fact that it controls the level of uncertainty in the approximate posteriors is extremely important (one of the selling points of the method is that the posterior doesn't collapse to a point).

It would be great to have some discussion of this earlier on, especially since it is essentially a dataset size.

Additionally, there's no discussion of whether or not the algorithm converges to anything and what that means.

One selling point of the population posterior by the authors is that since there's always model mismatch the posterior concentrating on a point in the parameter space is a bad thing.

But this statement seems to have the underlying assumption that people think that

their model is converging to the data generating distribution as more data arrives.

But I'm not certain people actually think this.

Having a fixed level of uncertainty (at least a lower-bound on it) through the \alpha parameter seems really useful for streaming data, I just don't think model mismatch is a good selling point.

The experiments section is well done and the experiments are convincing.

One question is whether some discussion can be made on why SVB does worse.

Is it local optima?

Additionally, the authors should state the actual step-size schedules that they used.

Are the results sensitive to the step-size schedule?

Lastly, how many replicates of permuting the order of the data did you use and can error bars be included?

The rest of my comments are minor:

- There are a lot of typos that need to be fixed.

- There is no future work in the "Discussion and Future Work" section.

Definitely include some because this is really interesting work.

I would like to reiterate that I thoroughly enjoyed this paper and the ideas it proposed.

I hope the authors address my concerns, especially those regarding clarity of presentation, and I think it would be a great addition to the proceedings.
Summary: This paper proposed an interesting method for Bayesian modeling of streaming data.

It would be a nice addition to the NIPS proceedings.

Submitted by Assigned_Reviewer_3

In this paper, population Bayesian inference is proposed for stream data.

By introducing population distribution, the authors try to increase model flexibility with population posterior. Stochastic population variational inference is proposed for model parameter learning.

Experimental results are reported in comparison with stream VB and stochastic VB.

There are several issues need to be addressed: 1) A more clear statement about the necessarily of population distribution is needed.

2) According to the paper, population VB should be able to capture the change of the data stream. But if  data points are from the current data set, what is difference between population

VB and SVI?

Why the sampling procedure for population VB can capture the current stream change if all data sample are treat equally?

3) With population distribution and parameter , we may get a more flexible model. But it comes with more computational cost due to the sampling procedure and additional parameter tuning.

Could the authors give a quantified computation time for all of the three methods on the data sets? Also details on how to choose .

4)

The reasons why population VB performs worse than stream VB on Twitter dataset.

Summary: Population Bayesian inference is proposed but lacks technical soundness.

Submitted by Assigned_Reviewer_4

- Summary of Paper

- The paper describes the development of an evidence lower bound

(ELBO) constructed by averaging over a data-generating

distribution. The paper shows that optimizing this ELBO leads to

impressive results on very large data streaming applications. - Quality

- L039: "the standard approach to probabilitic modeling", might be

better stated as "the standard approach to Bayesian probabilitic

modeling" since we can build a probabilitic model without a data

set in hand.

- L044: On initial reading, my reaction was, "Why shouldn't the

Bayesian posterior become more confident with more data? Indeed,

this is a desirable property of Bayesian inference procedures.

Also, "over-dispersion" is a well-known problem for many

generalized linear models and the typical solution there is to

build a better model. So, isn't the solution here, to build a

better model? Or if there is uncertainty about the model, perhaps

we should average over models in some way." However, later, some

clarity is provided in that the procedure aims to be robust to

model specification in a different way.

- L051: Again, my initial reaidng of this paragraph caused me to be

concerned that the real problem with Bayesian updating on data

streams is not the Bayesian procedure, but the way the model has

been specified. If the data stream is changing and we haven't

explicitely modeled that, then of course the updates may yeild

poor inferences, but that's not because our updating procedure is

flawed, but because our model is flawed. Here again, it seems

that the proposed procedure is trying to be robust to model

specification issues that really cause problem on data streams.

Perhaps the narrative in these introductory paragraphs can be

sharpened to set up the nice work presented later.

- L056: The claim is that explicitly modeling the time series

incurs a heavy inferential cost. Can this claim be supported with

a citation or other evidence?

- L165: Is there a misplaced parenthesis and perhaps a missing

\beta in the variantional distribution in the F-ELBO?

- L165: The standard ELBO is conditional on a particular data set x

and the F-ELBO is an average over data set x provided by the

population distribution X ~ F_\alpha. I'm curious if taking this

average causes the F-ELBO to preferentially optimize the ELBO

over modes of F_\alpha. Whereas, if we conditioned on a

particular x, as in the ELBO, it wouldn't matter how likely that

data set is under F_\alpha. Can the authors comment on the

tradeoffs of marginalizing over F_\alpha versus conditioning on a

sample from it?

- The results primarily deal with prediction rather than parameter

estimation. This is entirely appropriate given the applications

where streaming data is typically found. However, is there

anything that can be said about the parameter estimates,

especially given the first-order goal of maximizing the ELBO or

F-ELBO is to obtain parameter estimates?

- I do like that the F-ELBO explains SVI and provides a nice

framework for understanding that sampling procedure. But, I

wonder if one has in hand a generative model for p(X), what is

the costs/benefits of using that distribution as an averaging

distribution instead of X ~ F_\alpha? I understand that if our

model is misspecified, averaging with respect to that model may

exacerbate the updating problems outlined, and instead drawing

samples from F_\alpha is model-independent. Is there any guidance

as to another reason p(X) is a poor choice? - Clarity

- It would help to clarify exactly where the problems identified in

paragraph 2 and 3 in the introduction lie. L042 says that the

problems are with "Bayesian updating on data streams", but L044

says "the first problem is that Bayesian posteriors will become

overconfident" and L051 says "the data stream might change over

time". After reading these assertions several times, it becomes

clear what is intended, but I think the statement on L044 could

be better as "The first problem with Bayesian updating on data

streams is that Bayesian posteriors will become overconfident"

and L051 could be "The second problem with Bayesian updating on

data streams is that the data stream might change over time

causing ..." - Originality

- The paper is original and provides a good justification for SVI. - Significance

- I find the paper to be highly significant and I hope will be a

welcome addition in the community.
Summary: The paper describes an innovative way to handle inference in streaming data scenarios. Notwithstanding a few questions about the procedure, I find it a significant and important contribution to the community.

Submitted by Assigned_Reviewer_5

The authors propose a variational objective that minimises the KL to a "population posterior", that is the expectation of the usual posterior under an empirical distribution. They then use this formulation to derive a streaming variational algorithm where the objective is parameterised by an empirical distribution.

The introduction seems to claim that online Bayesian posteriors converge to a point mass: asymptotically, is this not correct and a consequence of consistency? Is the point the convergence can be premature? If so, how much of this is due to the variational (or other) posterior approximation which tends to underestimate uncertainty? i.e., is the problem really with Bayesian inference or with the approximation taken?

Eq (3): min -> argmin?
Summary: This is a very nice treatment of streaming Bayesian inference via variational methods. The experiments are strong, and the formalism is quite elegant.

Author Feedback
Author rebuttal: We thank the reviewers for their constructive feedback. We address their points below.

> [R2] According to the paper, population VB should be able to capture the change of the data stream. But if alpha data points are from the current data set, what is difference between population VB and SVI?

Thanks to the reviewer for this question. We mention at the end of Section 2 that SVI can be recovered from population VB with a finite dataset of size N using the plug-in principle [Efron & Tibshirani 1994] by setting \alpha=N and treating \hat{F}_X as the stream F. In general though, the two approaches are distinct: there is no "current data set" in population VB and we are free to set \alpha. We will make this clearer in the paper, and include more detail about \alpha (as per comments by other reviewers).

> [R2] Why the sampling procedure for population VB can capture the current stream change if all data sample are treat equally?

When we use classical Bayesian inference on a data stream (or an approximation based on Bayesian updating, e.g., streaming VB) posterior concentration leads data points earlier in the stream to have more effect than data points later in the stream. E.g., if the stream goes from F_0 to F_1 after 1M data points, it will take more than 1M data points to capture F_1. Population VB addresses this issue by introducing \alpha, the effective sample size (in the example, \alpha can be set to 100K).

> [R2] A more clear statement about the necessity of population distribution is needed.

The population distribution is the data generating mechanism behind all the data seen in the stream, of which there may be an infinite number. This provides a simple way to adapt existing models to streaming settings (both stationary streams and non-stationary streams).

> [R2] The reasons why population VB performs worse than stream VB on Twitter dataset.

SVB washes out the prior faster than gradient-based approaches (SVI, population VB), which makes it more sensitive to local optima. In some cases, however, this is fortuitous. Twitter data is evidently one such case, possibly because we must infer local topic assignments from the very limited evidence of each tweet.

> [R2 asks about computational complexity of the three methods]

The dominating computational cost for all three methods is to fit the local variational parameters. The point of departure is in the way they update the global parameters. Our experiments took a few hours to pass through millions of documents on a consumer laptop; we will provide more details about this in the next version of the paper.

> [R2] details on how to choose alpha.

In the beginning of the empirical evaluation section we mention that we found empirically that the best \alpha is often different to the actual number of distinct data points (see Figure 3). This supports the argument that model mismatch is a problem. Lower values of \alpha are preferred when greater model mismatch is anticipated.

> [R3 asks why time series models incur a heavy cost and R4 comments that we did not compare to time series models].

Time series models are non-exchangeable, often requiring more iterations during inference (e.g., VB or MCMC) because the Markov blanket of time dependent random variables grows to include other time-dependent variables to which they are usually correlated. We reiterate that population VB also works in the stationary case, where time series are not appropriate.

We agree that a direct empirical comparison to time series model equivalents would shed light on how much we lose from the exchangeability assumption and how much we can gain back by considering the population posterior. However, work on large scale time series modeling (the order of data sets we consider here) is still an active area of research.

> [R1] there's no discussion of whether or not the algorithm converges to anything and what that means.

In general, if the gradients are unbiased, which they are for population VB, then an appropriate step size schedule will ensure convergence to a local optimum by stochastic optimization theory [Bottou 1998]. We will make this explicit in the paper. Thanks for the suggestion.

> [R1 says it is confusing what exactly the F-ELBO is a proxy objective for near l. 153]

Maximizing the F-ELBO minimizes the *expected* KL between the approximate and actual posterior. This quantity is an upper bound on the population posterior. We will add another sentence to the paragraph to avoid confusion.

> [R1] The title should really use "Bayesian modeling" instead.

We thank the reviewer for the excellent suggestion. We will change the title.

> [R5] Eq (3): min -> argmin?

Thanks for bringing this mistake to our attention; we will fix it. We will also fix the typos mentioned by the other reviewers.